# Effects of Enrichment Type, Presentation and Social Status on Enrichment Use and Behavior of Sows—Part 2: Free Access Stall Feeding

**DOI:** 10.3390/ani12141768

**Published:** 2022-07-10

**Authors:** Cyril Roy, Victoria Kyeiwaa, Karen F. Mancera, Yolande M. Seddon, Laurie M. Connor, Jennifer A. Brown

**Affiliations:** 1Prairie Swine Centre, 2105-8th Street East, P.O. Box 21057, Saskatoon, SK S7H 5N9, Canada; cyrilroy72@gmail.com (C.R.); victoria.kyeiwaa@usask.ca (V.K.); karen.mancera@usask.ca (K.F.M.); 2Department of Animal and Poultry Science, University of Saskatchewan, 51 Campus Drive, Saskatoon, SK S7N 5B4, Canada; 3Western College of Veterinary Medicine, University of Saskatchewan, 52 Campus Drive, Saskatoon, SK S7N 5B4, Canada; yolande.seddon@usask.ca; 4Department of Animal Science, Faculty of Agricultural and Food Sciences, University of Manitoba, Winnipeg, 201-12 Dafoe Road, MB R3T 2N2, Canada; laurie.connor@umanitoba.ca

**Keywords:** environmental enrichment, social status, sows, aggression, habituation

## Abstract

**Simple Summary:**

The Canadian pork industry is transitioning to group housing gestation. The Code of Practice for pigs recommends the provision of enrichment. Although straw is considered prime enrichment, it is perceived as a biosecurity risk and manure management issue; therefore, identifying other enrichment materials is needed to promote adequate welfare. Previously, our group examined the effects of four types of point-source enrichments in an Electronic Sow Feeding (ESF) system. This study is a continuation of our previous research applied in Free Access Stalls (FAS). Four treatments were studied: (1) Constant: constant provision of wood on chain; (2) Rotate: rotation of three enrichments (rope, straw, and wood on chain); (3) Stimulus: rotation of three enrichments with an associative stimulus (bell or whistle); and (4) Control: no enrichment. Contact with enrichment and time spent in different postures were measured using scan sampling. Skin lesions were scored, and salivary cortisol was measured in a subset of dominant and subordinate sows. Sows spent more time contacting straw compared to other types of enrichment. There was no difference in enrichment use between dominant and subordinate sows, no increase in cortisol concentrations in subordinates nor significant differences in aggression. In conclusion, sows preferred straw enrichment and FAS provided some protection against aggression and stress.

**Abstract:**

Continuing with previous research by our group in an ESF system, four types of enrichment treatments were assessed in gestating sows housed in Free Access Stalls: (1) Constant: constant provision of wood on chain; (2) Rotate: rotation of rope, straw and wood; (3) Stimulus: rotation of enrichments with an acoustic cue; and (4) Control: no enrichment. Treatments had a 12 day-duration. Four groups (28 ± 2 sows) were studied from weeks 6 to 14 of gestation. Groups received all treatments in random order. Three dominant and 3 subordinates per pen were selected using a feed competition test. Digital photos were collected at 10 min intervals for 8 h on days 1, 8, 10 and 12 to record interactions with enrichment. Skin lesions were assessed on days 1 and 12, and salivary cortisol was assessed in weeks 6, 10 and 14 of gestation. More enrichment use was observed in Rotate and Stimulus treatments compared to Constant, and more sows contacted enrichment when straw was provided in the Rotate and Stimulus treatments. There was no difference in the amount of enrichment use by dominants and subordinates, no cortisol concentration elevation in subordinate sows nor any difference in lesion scores. In conclusion, social status had little impact and feeding system is important to reduce stress and aggression.

## 1. Introduction

Group housing is becoming more common on North American farms. Sows in these systems have welfare benefits such as increased freedom to move and social enrichment through interaction with other sows. However, the success of group housing depends on space availability and the presence of an appropriate feeding system. An added advantage of group housing is the increased opportunity to introduce enrichment and improve pig health and welfare [1]. Several studies indicate that enrichment increases social and overall activity [2,3]. In growing pigs, enrichment has been found to increase adaptability to novelty and learning capabilities, as well as reduce tail and ear biting, belly nosing, and fear [1,4,5,6,7,8,9].

In North America, the use of substrate enrichments is generally avoided due to their potential for blocking the slurry system associated with slatted floors [4]. In contrast, enrichment materials that are fixed to specific positions in the pen (point source object enrichment) can be used successfully in pens with slatted floors. Some materials that have been studied are rubber toys, pieces of wood, chains, and even garden hoses [10,11,12,13] as well as straw or hay [1,5,10] and the use of pheromones [6].

Enrichment is affected by many variables. For instance, its effectiveness depends on its cleanliness, accessibility, attractiveness, deformability, destructibility, and immobility to ease of manipulation [12]. Likewise, novelty related to material presentation increases attraction to the materials [14] and creating anticipation for the enrichment through a sound stimulus generates an increase in play and reduced aggression and lesions in piglets following weaning [7]. Social rank also interferes with enrichment use; if a material is considered valuable and the access is limited, dominant individuals will displace subordinates and restrict enrichment use [15].

Another factor influencing enrichment use is habituation, which has been shown to greatly affect point source enrichments [10,11,16], as animals are less interested if the same material is repeatedly used [17]. Interaction with point source enrichment is high in the first 24 h and reduces over time [6]. Furthermore, habituation is also affected by the material properties and presentation [6]. For instance, when sisal rope was offered hanging inside the pen, there was a higher initial response and interaction compared to concrete blocks and a rubber boot on the first day [16]. Likewise, enrichments that are deformable, chewable and destructible are more attractive to pigs [11,12,16].

Another factor that impacts enrichment use by sows is feeding management. Two common feeding management systems in group housing are Electronic Sow Feeding (ESF) and Free Access Stalls (FAS) [18]. It is likely that different feeding systems influence enrichment use in different ways to impact sow welfare, particularly in relation to aggression [10,18,19,20] and health risks such as lameness, injury and abortion [21]. A previous study by Roy et al. [10] focused on the effects of enrichment in an ESF system; when enrichment was provided, the level of enrichment use was similar for dominant and subordinates sows, but subordinate sows showed greater general activity levels, increased injury scores and higher salivary cortisol levels. Since FAS provide more protection to subordinate sows than ESF, it is possible that aggression is mitigated in FAS, influencing enrichment interaction.

Given this information, it is important to generate more information on how enrichment use and habituation are influenced by different materials and presentation schemes, in order to improve enrichment implementation in group housing. Previously, our group investigated how enrichment type, method of presentation and social rank affected enrichment use and sow behavior in an ESF system [10]. We hypothesized that enrichment rotation would increase enrichment use and reduce habituation, and that the use of an associative stimulus would increase the initial response to enrichment provision. Our hypotheses were confirmed when an increased enrichment contact was found if different enrichment materials were rotated and when a bell or whistle was used before enrichment presentation; however, subordinate sows received more skin lesions and had higher salivary cortisol levels [10]. Due to the high competitiveness present in ESF systems, it is possible that the feeding system can also influence enrichment use and habituation in group housing. Therefore, as a direct continuation of our previous enrichment research in an ESF system, in this study we explore how enrichment use and habituation are influenced by a FAS feeding system. We hypothesized that enrichment use in FAS would differ from ESF by reducing social stress and aggression among sows.

## 2. Materials and Methods

This research was conducted at the Prairie Swine Centre’s research barn associated with University of Saskatchewan, in Saskatoon, SK, Canada. The experiment was approved by University of Saskatchewan Animal Research and Ethics Board (AUP#20140037) and adhered to the Canadian Council on Animal Care guidelines for humane animal use.

### 2.1. Animals and Housing

A total of 224 sows (28 sows/pen in 8 replicates, PIC Camborough × Line 337) were studied. Sows were housed in four gestation pens with partially slatted concrete floors and free access stalls for feeding (Egebjerg INN-O-STALL^®^ free access stalls, Egebjerg International A/S, Nykøbing Sjælland, Denmark). Each feeding stall had a nipple drinker and a single space feeder located at the front. Each gestation pen consisted of an ‘I-area’ with concrete slatted flooring (slot width 30 mm, slat width 120 mm) running between two rows of sixteen free-access stalls (0.65 m × 2.1 m) and a solid floored T-area/loafing area (3.05 m × 7.3 m) at one end of the pen (Figure 1). Sows also had access to one nipple drinker in the loafing area. Water was given ad libitum through the drinkers. Sows were provided 2.3 kg of a standard gestation diet once a day at 7:00 a.m. Sows were artificially inseminated with pooled semen and remained in breeding stalls for four to five weeks. They were moved to gestation pens every 2 weeks, thus at 4- or 5-weeks post-breeding, and trials started one week after mixing. Routine health checks were conducted daily by production staff.

Three dominant and 3 subordinate sows per treatment (*n* = 48) were marked for identification in week 1, using feed competition tests, following Anderson et al. [21]. Briefly, feed competition tests were performed after the initial aggression associated with mixing had resolved (on days 3–5 after mixing). The solid floor area was scraped and 4 kg of feed were poured onto the floor in two lines (2 kg per line, approx. 1 m long, lines separated by at least 2 m). Once feed was poured, sows competed for access. On day 5, the three sows that gained first access to feed were identified as ‘dominant’ (Dom), and three sows that did not participate or were driven away were identified as ‘subordinate’ (Sub).

### 2.2. Treatments

Treatments groups in this study are based on those set by Roy et al. [10], as this study is a continuation of the results obtained in our previous study in an ESF system. Briefly, treatments were provided to each group for a period of 12 days, in a randomized fashion within group with a two-day interval between each treatment. The treatments consisted of:Constant: provision of one type of enrichment—wood on chains (3 per pen).Rotate: rotation of three enrichments—rope, straw, wood on chain (3 per pen).Stimulus: rotation of three enrichments (as described for Rotate) with an associative stimulus used to signal the arrival of enrichment. The associative stimulus used was a bell or whistle (duration: 2 s), and was switched half-way through the study so that any sows that returned to the study (in their next gestation) would not be familiar with the stimulus, andControl: no enrichment.

As described by Roy et al. [10], for the Rotate and Stimulus treatments rope was presented on Mondays, straw on Wednesdays and wood on Fridays (Figure 2). Additionally, for the Stimulus treatment, a bell or whistle was used just before enrichment was provided to generate an associative stimulus. Wood on chain and rope were hung over the solid floored area with snap hooks to attach them to chains suspended from a metal pole. The rope was 1.2 m long, with a 15 cm tassel at the end, using three-stranded cotton ropes of 19 mm in diameter, and suspended 20 cm above floor level. Straw was provided on the solid floor area (300 g/sow; 8.4 kg in total). Sows consumed most of the straw, but if any remained, it was removed before placing the next enrichment. The wood enrichment was made of softwood (5 cm × 10 cm, 1.2 m long) and hung from the chain to rest in the floor at a 45° angle.

### 2.3. Data Collection

Using the same methodology stated in Roy et al. [10], sows were weighed, their parities recorded and moved to group pens at 4–5 weeks of gestation.

One camera was mounted over each pen to record sow activity and enrichment use. Digital photos were taken at 10-min intervals over 8 h per day from 8 a.m.–4 p.m. on days 1, 8, 10 and 12 for each treatment (48 photos/day; 192 total). A trained observer determined the location and posture of all sows visible in the photos of the enrichment area. The number of sows standing, lying, or sitting at each time point was recorded, taking into account only those sows that were observed clearly. An ethogram of the behavior categories recorded is shown in Table 1. For the Rotation and Stimulus treatments, recordings were performed within the first 8 h after providing new enrichments. In the Stimulus treatment (a bell or whistle) was played for 2 s as soon as cameras were started and was followed by enrichment provision. Enrichment use was studied for the whole group and for the six selected sows (Dom and Sub) using the digital photos. Dom and Sub sows were identified by blue (Dom) or red (Sub) spray markings.

#### 2.3.1. Skin Lesion Assessment

To evaluate levels of aggression in pigs, lesion scores were used as described in Roy et al. [10]. Briefly, using methods adapted from Hodgkiss et al. [22], skin lesion scores were assessed on day 1 (before enrichment was provided) and on day 12. Lesion scores ranged from 0 (no injury) to 3 (severe injury), and were assessed on 11 regions (head, ear, neck, shoulder, top of back, tail, vulva, hind leg, side, udder and front leg) on both sides of the body (Figure 3). Counting fresh injuries only, the score was as follows: 0 = No injury (skin unmarked: no evidence of injury), 1 = Slight injury (<5 superficial wounds), 2 = Obvious injury (5–10 superficial wounds and/or <3 deep wounds), 3 = Severe injury (>10 superficial wounds, and/or >3 deep wounds).

#### 2.3.2. Cortisol in Saliva

As described in Roy et al. [10], saliva samples were taken from Dom and Sub sows per group (*n* = 48) at the end of week 1 after the first treatment, at day 11 of the second treatment (i.e., week 9 of gestation), and when the final treatment was completed (week 14). Saliva samples were collected between 8 and 9 a.m. each day to control for diurnal variation in cortisol levels. To collect saliva, sows to chew on large cotton buds thoroughly moistened (30–60 s/sample). The buds were placed in centrifuge tubes (Fisher Scientific, Ottawa, ON, Canada) and centrifuged immediately for 15 min at 830× *g* (Beckman TJ-6 Centrifuge, Beckman Coulter, Mississauga, ON, Canada) to remove mucins and other particulate matter. Saliva samples were transferred to labeled storage tubes using disposable pipettes and stored at −20 °C until analysis.

The Salimetrics^®^ Cortisol Enzyme Immunoassay Kits (Salimetrics, State College, PA, USA) were used to assess the cortisol concentration in saliva. The kit is a competitive immunoassay designed for quantitative measurement of salivary cortisol using a 96-well ELISA plate (Salimetrics, State College, PA, USA) with spectrophotometric detection at 450 nm. Each 96-well plate contained six standards, one zero and one nonspecific binding sample, two controls and 38 samples, which were run in duplicate. Intra-assay precision estimates on low and high standards (*n* = 20) gave values of 1.14 ± 0.05 and 0.16 ± 0.01 µg/dL (mean ± SD) and CV’s of 4 and 5%, respectively. Inter-assay precision estimates on low and high standards (*n* = 20) gave values of 1.14 ± 0.05 and 0.18 ± 0.01 µg/dL (mean ± SD) and CV’s of 4 and 9%, respectively.

### 2.4. Statistical Analysis

Data manipulation and analysis were performed using the models and procedures described by Roy et al. [10].

Data manipulation: Using the pen as the experimental unit, outcome variables associated with enrichment use (location and postures) both for group level and focal sow photo scan observations were analyzed. The proportion of time spent performing each behavior (defined in Table 1) was calculated as the proportion of scans (photos) where the behavior was observed out of the total number of scans (photos) recorded during the 8-h observation period (48 scans per day). The number of sows performing each behavior was calculated by the average proportion of sows that performed the behavior when their behavior was observed.

Sow was the experimental unit for lesion scores and cortisol analysis. For lesion scores, a combined skin lesion score was created by summing the lesion scores for all body regions. The effects of different enrichment presentations and treatment day (days 1 and 11) were assessed using the total body score. To study lesion scores in different body regions, the 11 regions (Figure 3) were condensed into four regions (head, shoulder, side and hind) and assessed for associations with independent variables enrichment treatment, social status and parity. Parity was categorized in three groups: parity code 1 (parities 1 and 2), parity code 2 (parity 3) and parity code 3 (parities 4 to 7).

Data Analysis: Contact with enrichment, location of sows and postural behavior data were analyzed using GLMM with beta regression models to account for proportional data in SAS 9.4 (SAS Institute Inc., Cary, NC, USA). Model fit was assessed by plotting the residuals. Fixed effects in the group behavior model included enrichment treatment, day of treatment (day 1, 8, 10 and 12) and their interaction, if significant. For focal sows, the effect of social status on sow behavior was studied in a similar model with social status added as a fixed effect and including two-way interactions of social status and treatment and social status and day of treatment. Interactions were removed from the model when not significant. Sow group (replicate) and day of treatment were used as random effects to account for the correlation associated with repeated measurement of the same animals. Day of gestation was controlled for as each group (pen of sows) received all treatments, and the order of enrichment treatments was randomized. The significance level was set at *p* < 0.05.

A mixed model (Proc Mixed, SAS 9.4, SAS Institute Inc., Cary, NC, USA) was used to assess the association between cortisol concentration and fixed effects of social status and stage of gestation, with sow group (replicate) as the random effect. Assay values obtained in µ/dL were converted to nmol/litre using the conversion factor 27.59. A similar model was used for lesion scores, with measures for day 1 and 12 observations being analyzed separately, and individual sow ID was added as a repeated measure. Cortisol and lesion values were log transformed to achieve normality.

## 3. Results

This study is a direct continuation of previous research performed by our research group. The results section has been purposefully summarized results in a similar fashion to our previous study in a ESF system [10], making tables and graphs comparable to our previous results to facilitate the identification of differences in enrichment use and habituation in ESF and FAS systems.

### 3.1. Group Level Observations

#### 3.1.1. Enrichment Use at Group Level

The proportion of time sows were in contact with the enrichment was significantly affected by treatment. Sows in the Rotate and Stimulus treatments spent more time in contact and in proximity to enrichments (less than 1 m), compared to the Constant treatment (Table 2; Contact: *p* = 0.021, Proximity: *p* = 0.001). The proportion of time sows were in contact with the enrichment was also affected by the type of enrichment offered. On day 10, when straw was provided in the Rotate and Stimulus treatments, sows contacted enrichment more frequently in these treatments compared to rope (days 1 and 8) or wood (day 12; Figure 4). Likewise, a greater proportion of sows in the Rotate and Stimulus treatments contacted the enrichment when straw was provided (Figure 5).

#### 3.1.2. Behavior at Group Level

When sow postures (standing, sitting, and lying) were studied at group level, a significantly greater proportion of sows spent time standing in the Constant, Rotation and Stimulus treatments compared to Control (*p* = 0.003, Table 3).

### 3.2. Focal Pig Observations

The average parity of Dom and Sub sows (i.e., actual parity, not parity code) was 3.11 ± 0.09 and 2.89 ± 0.08, respectively. There were more parity 1 sows with Sub social status than with Dom. Sows’ body weight increased significantly with parity, resulting in Dom sows being significantly heavier than Sub sows (Dom = 256.49 ± 4.24 kg; Sub = 239.3 ± 4.24 kg; LS Means ± SEM).

#### 3.2.1. Enrichment Use by Focal Sows

There was no difference between Dom and Sub sows with respect to the time sows spent in contact with enrichment, the proportion of sows in contact with enrichment or their proximity to the enrichment (Table 4). Day had a significant effect, with a greater proportion of sows and time spent in contact with the enrichment on day 10, when straw was provided (*p* < 0.02, Table 4).

Similar to the group level results, significant treatment effects were found for enrichment use and proximity in focal sows. Focal sows spent more time in contact and in proximity to the enrichment in the Rotate and Stimulus treatments than in Constant. Likewise, a greater proportion of sows were observed in contact with the enrichment in the Rotate and Stimulus treatments (Table 5).

#### 3.2.2. Behavior of Focal Sows

Social status did not influence standing or lying behavior, but had a significant effect on the proportion of time sows spent sitting. Dom sows spent more time in this position and there was a tendency for a higher proportion of Dom sows performing this behavior (Table 6; *p* = 0.058). No interactions were found among social status, treatment, or day for sitting behavior.

#### 3.2.3. Skin Lesion Scores

There was no effect of social status on skin lesions (Table 7). The effect of parity (parity code) was also non-significant. Enrichment treatment significantly affected the number of shoulder lesions on day 1, with the Constant treatment presenting the highest shoulder lesion score, the Rotate the lowest, and with the Control and Stimulus treatments being intermediate (Table 7). Additionally, treatment tended to affect the total skin lesion score on day 1 (*p* = 0.065; Table 7). There was a significant effect of treatment on total lesion scores on day 12, with the Stimulus treatment presenting the highest average score and the Constant treatment presenting the lowest.

#### 3.2.4. Cortisol Level

Social status did not affect focal sows’ saliva cortisol levels. However, there was a significant change over time; samples collected in early and mid-gestation had significantly lower cortisol values than samples collected in late gestation (Table 8).

## 4. Discussion

This study compared how different types of enrichments and the way they are presented as well as social status, shape enrichment interaction and sow behavior in a FAS feeding system. A previous study by Roy et al. [10] evaluated the same treatments and variables in an ESF system. This study is a direct extension of these results in a FAS system. We considered it important to compare enrichment use in different feeding systems because feed accessibility and pen design play a role on determining priorities in terms of resource availability, daily activity and levels of stress.

To compare how proximity and time spent with the enrichment were affected, straw, rope and wood were presented in two treatments (rotate and stimulus). As straw, being a consumable enrichment, is preferred over non-consumable enrichments, we took this material as a positive control to compare different enrichment materials [22]. Our results indicate that, similar to ESF and other enrichment studies [2,4,10,23,24], animals had more contact with the straw compared to point source wood and rope enrichments and overall presented a greater proportion of sows standing compared to non-enriched controls, suggesting greater behavioral activity when enrichment is provided. Straw was placed on the solid floor area, thus making it more accessible compared to hanging rope or wood which could have influenced its greater use. Providing 300 g/sow in partially slatted flooring pens did not block the manure system because most of the material was ingested and only a small amount entered the manure pits. This observation is valuable information for use of this material under commercial conditions.

Rope and wood were not contacted as often as the straw, but they are still valuable enrichments to be considered. Previous studies have reported ‘hanging rope’ as an effective enrichment strategy [4,10,13,15]. Multiple studies indicate that some of the most important enrichment characteristics are ingestibility, flexibility and ability to elicit sustained interest [4,11,12]. Rope and wood are flexible and destructible and can complement the provision of enrichment, but more desirable materials may also be needed, since limited access to highly sought objects may increase competition and aggression [25]. Therefore, the types and optimal ratio of enrichment objects provided to sows should be further evaluated.

Enrichment interest can be stimulated by changing and renewing materials regularly to reduce habituation [11,26]. We postulated that frequent rotation of materials would maintain novelty, diminish habituation, and enhance sows’ activity levels. In the Rotate and Stimulus treatments, rotating enrichments resulted in significantly more sows in contact and in proximity to the materials (within 1 m). As observed in ESF [10], rotating enrichments every three to four days was a successful strategy to promote interaction. Further studies should evaluate the effect of different rotation frequencies and delays to re-exposure on sow behavior and enrichment use.

No increased initial response was found in the Stimulus treatment. Initiating enrichment provision concomitantly with an auditory cue has been used to increase its value. For instance, cognitive performance was improved in gilts when straw was provided in addition to food rewards (mixed seeds or sliced apples) after an auditory stimulus [7,27]. These studies intensively trained pigs to learn the association between the nutritional enrichment and the auditory stimulus. In this study we aimed to simulate enrichment provision in commercial practice, so training was not included. Thus, we are unable to confirm if sows formed an association between the sound and the enrichment. Future research should include training so sows are able to learn the correct association. The possibility of an associative stimulus increasing aggression if enrichment accessibility is limited should be acknowledged.

Providing enrichment increases activity, represented by standing behavior [28,29,30]. In this study, enrichment provision (Constant, Rotate and Stimulus treatments) was associated with more sows standing, suggesting a greater activity level. Gestating sows spend most of their time lying (roughly 80% of total time) [31]. Still, moderate activity levels are positive, as seen in research contrasting sows in groups and kept in stalls [32]. Further studies should investigate the benefits of enhanced activity and define what are ‘appropriate’ activity levels for gestating sows, to better define and measure the desired outcomes of enrichment provision.

In terms of social status, we theorized that Dom sows would have more access to enrichment and/or at better times of the day if enrichment was regarded as a valuable resource, compared to Subs in a FAS feeding system. However, no effect of social status was found, and the average proportion of Dom and Sub sows contacting enrichment was approximately 35%, indicating that when sows were observed, typically one individual was present. In the previous enrichment study using ESF, Sub sows spend more time near enrichments compared to Dom sows [10]. The ESF requires sows to feed in sequence, such that Dom sows will compete to obtain access and guard the feeder [33,34], affecting enrichment use. The FAS system provides simultaneous daily feedings and eliminates competition at the feeder, which may have resulted in Dom and Sub sows having equal opportunity to contact the enrichments. Our results are consistent with previous findings, as social rank did not significantly affect the motivation of sows in stalls to access a group pen with enrichment [15]. In the study, Dom and Subs sows fed in stalls showed similar levels of operant response and latency when pressing a panel to access enrichment objects [15].

Elmore et al. [15] also observed that social status had a significant effect on activity, with Dom sows presenting more activity and standing more compared to Sub sows. In this study, Dom and Sub sows had similar standing behavior, and Dom sows spent more time sitting compared to Sub sows. The behavioral observations were collected with a camera pointing to the enrichments; thus, the behavior of Sub and Dom sows in other pen areas was not observed. The similar enrichment use and activity levels could suggest that there were sufficient enrichment objects available to avoid competition (three per 28 sows), or that Dom sows did not perceive the enrichments as highly valuable. Further studies should evaluate other pen areas and explore variation in the number of enrichment objects available per sow.

Lesion scores are useful to determine the aggression levels in pigs [35]. Severity, amount of time and frequency of aggressive events can be assessed by the number of injuries on pigs, especially those on the anterior region of the body [35,36]. Aggression is influenced by familiarity of pigs, space, group size and composition, pen design, time of day, food and type of bedding [37]. Aggression is also affected by resource competition. For instance, when group-housed grower-finisher pigs were given limited access to straw, an increase in aggression was observed [26]. Thus, we postulated that Sub sows would receive more skin lesions than Dom; however, there was no difference in lesion scores. Injury scores found in this study were also lower compared to those observed in our previous study in an ESF [10]. As mentioned before, sows in this study were fed once a day in FAS, reducing competition over feed and aggression between Sub and Dom sows, indicating that this management system is more likely to reduce the prevalence of injuries.

Enrichment treatments affected lesion scores, with sows in the Constant treatment receiving more lesions than any other treatment, particularly in the shoulder region on day 1 of enrichment provision. Because lesions in day 1 were the baseline before enrichment provision, differences cannot be attributed to enrichment. Sows could have fought more during the two-day time interval between treatments, when sows had no enrichments. In contrast, there was an increase on lesion scores on day 12 for the Stimulus treatment. This finding contradicts previous research, in which the announcement of enrichment (straw and mixed seeds) by an acoustic cue (doorbell with a sound pressure of 80 dB at 1 m distance) reduced aggressive behavior and injuries compared to enrichment alone in weaned piglets [7]. In this regard, it has been proposed that acoustic cues reduce stress by increasing predictability and controllability of events [38]. For example, increasing the predictability of food arrival using a bell decreased the performance of agonistic behavior in pigs [39]. However, in this study, although the whistle announced enrichment items, materials were different each time, which may have decreased environmental predictability and increased aggression and lesion scores for sows in the Stimulus treatment. Furthermore, although associating an acoustic cue with the provision of enrichment can be beneficial, it is important to consider the type of sound and the volume, since inappropriate use of acoustic cues can backfire and even have negative effects [40].

Cortisol concentrations in Sub sows and Dom sows did not differ in this study. Since animals received all enrichments randomly and collection of saliva occurred only at three time points during gestation, the effects of enrichment on salivary cortisol cannot be determined. In group gestation, cortisol levels are related to social aggression, rising significantly in the 24–48 h after mixing, and returning to baseline after a period of 48 h [41]. If competition over resources is absent, difference in cortisol levels between Sub and Dom sows after one week should be absent [42]. However, if resources are insufficient (space, food or environmental enrichment) stress levels will remain high [43] particularly in Sub sows. In this study, the FAS feeding system may have reduced aggression and contributed to an overall reduction in stress. In comparison, when enrichment use was studied in ESF [10] the levels of salivary cortisol were greater for Sub sows compared to Dom, possibly because of competition related to feeder access; furthermore, mean salivary cortisol levels in ESF were over 1.5 × higher than those observed in this study, indicating a potential effect of feeding system on the generation of stress. Finally, sows had higher levels of salivary cortisol in week 14 of gestation compared to earlier samples. This increase during late gestation is expected, since sows (and other species) are known to experience an increase in cortisol levels when approaching parturition [44,45,46].

## 5. Conclusions

Enrichment and how it was presented affected significantly the number of sows contacting the enrichment materials. Straw produced the greatest response without affecting the manure system. Social hierarchy (dominant or subordinate) did not have a significant effect on enrichment usage which highlights the importance of the type of feeding system on the behavioral response. Subordinate sows in this study did not show an increase in skin lesions or salivary cortisol concentrations compared to dominants, suggesting that FAS reduces competition for feed and enrichment objects, reducing the risk of aggression on subordinates, in contrast to ESF systems where the same type of enrichments and presentation produced more lesions in subordinate sows.

## Figures and Tables

**Figure 1 animals-12-01768-f001:**
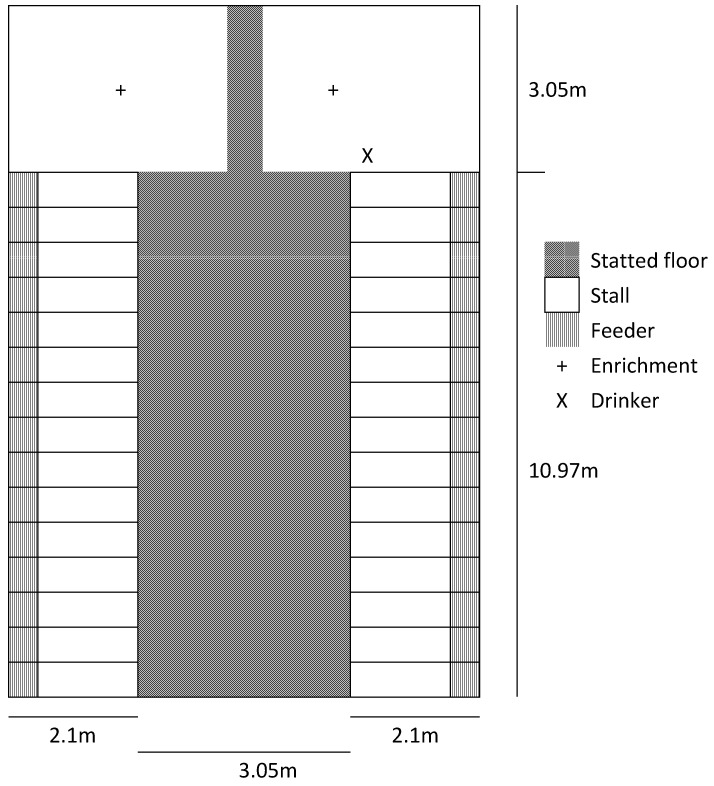
Schematic diagram of a gestation pen.

**Figure 2 animals-12-01768-f002:**
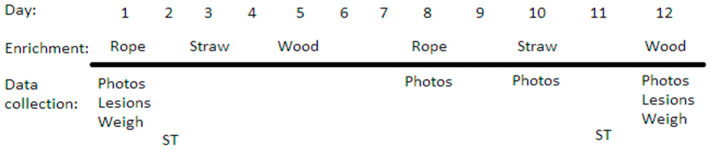
Timeline illustrating data collection and the rotation schedule for enrichments in Rotation and Stimulus treatments (ST). Each treatment lasted 12 days, followed by two days off. Four treatments were provided consecutively to each pen group in random order over a period of eight weeks (beginning at 5–6 weeks and ending at 13–14 weeks of gestation). Adapted from Roy et al. [10].

**Figure 3 animals-12-01768-f003:**
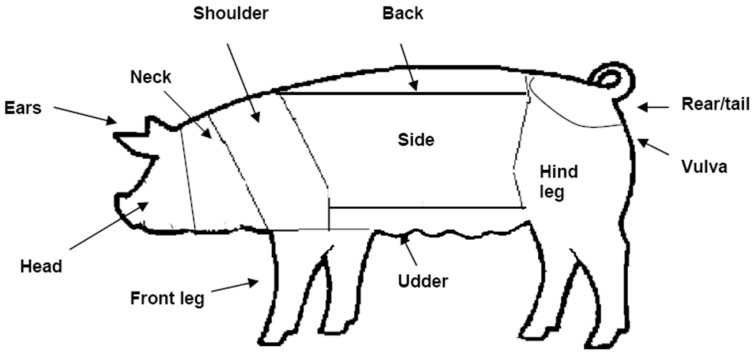
The animal’s body was divided in to 11 areas as illustrated. Each area on both sides was assessed for skin lesions (score = 0 to 3). Adopted from Roy et al. [10].

**Figure 4 animals-12-01768-f004:**
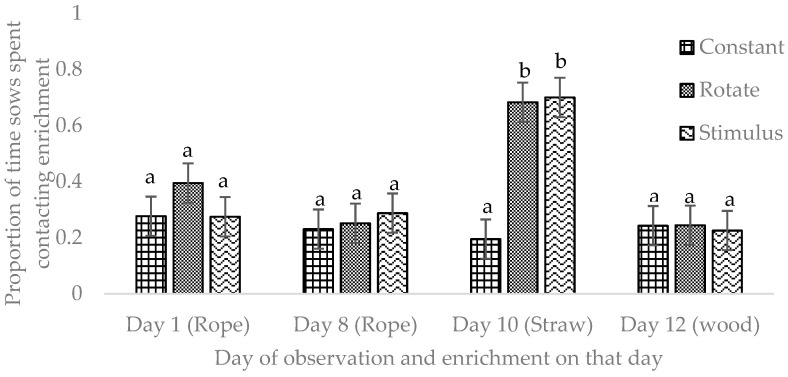
Proportion of time sows contacted enrichments (rope, wood or straw) and day of observation (days 1, 8, 10 and 12) in the Constant, Rotate and Stimulus treatments. Proportion of time was calculated as the proportions of scans (time lapse photos at 10-min intervals) where the behavior was observed out of the total number of scans during an 8-h period (8:00–16:00 h) on four days (1, 8, 10 and 12). Note: sows in the Constant treatment received the wood enrichment on all days (*n* = 46, F = 2.95, *p* < 0.029). ^a,b^ Bars without a common superscript are significantly different at *p* < 0.05.

**Figure 5 animals-12-01768-f005:**
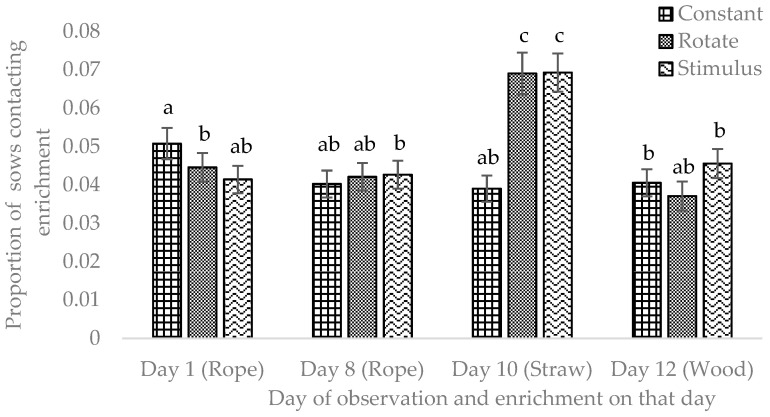
Proportion of sows contacting enrichments (rope, wood or straw) and day of observation (days 1, 8, 10 and 12) in the Constant, Rotate and Stimulus treatments. Note: sows in the Constant treatment received the wood enrichment on all days. Sows were observed during an 8-h period (8:00–16:00 h) on four days (1, 8, 10 and 12) using time lapse photos taken at 10-min intervals (*n* = 46, F = 7.72, *p* < 0.001). ^a,b,c^ Bars without a common superscript are significantly different at *p* < 0.05.

**Table 1 animals-12-01768-t001:** Ethogram used to identify sow posture and location relative to enrichment. Adopted from Roy et al. [10].

Location/Posture	Definition of Location or Posture
In contact	Sow’s snout is in contact with enrichment. For straw enrichment, the sow should be standing on or facing the straw and appears to be contacting the material with her snout.
Less than 1 M	Sow is not in contact with enrichment, but head is within approximately 1 m of enrichment. For straw enrichment, the sow is lying or standing in proximity to the straw (head is <1 m from straw).
Greater than 1 M	Sow’s head is greater than 1 m from enrichment. Includes all visible sows in the photo that are not in contact or <1 m.
Standing	Sow is upright on four legs (not sitting or lying). If the sow is in the process of lying down, however still upright, and rear end is supported on hind legs, she is considered standing.
Sitting	Hind end is in contact with the floor; front end is raised and supported by front legs.
Lying	Sow is lying (ventral or lateral). In ventral lying, the sow’s belly region is in contact with the floor. In lateral lying, one side of the body is in contact with the floor and legs are extended to one side.

**Table 2 animals-12-01768-t002:** Effects of enrichment treatment, day of observation and their interaction on time spent (proportion of observations) and number of sows (proportion of sows) in contact with enrichment, less than 1 M or greater than 1 M from enrichment.

Behavior *	Treatment			SEM	*p*-Value		
	Constant	Rotate	Stimulus		Treatment	Day **	Treatmentx Day
Contacting enrichment							
Time (prop. of obs.)	0.235 ^a^	0.385 ^b^	0.362 ^b^	0.041	0.021	0.009	0.029
No. of sows (prop.)	0.042 ^a^	0.047 ^b^	0.049 ^b^	0.003	0.024	<0.001	<0.001
Less than 1 M							
Time (prop. of obs.)	0.169 ^a^	0.495 ^b^	0.296 ^b^	0.046	0.001	0.001	<0.001
No. of sows (prop.)	0.042 ^a^	0.053 ^b^	0.051 ^b^	0.002	0.032	<0.001	<0.001
Greater than 1 M							
Time (prop. of obs.)	0.991 ^a^	0.971 ^a^	0.858 ^b^	0.015	0.015	0.167	NA ***
No. of sows (prop.)	0.124	0.125	0.125	0.008	0.807	0.003	0.018

^a,b^ LSMeans with different a superscript within the same row are significantly different (*p* < 0.05). * Sows were observed during an 8 h period (8:00–16:00 h) on four days (1, 8, 10 and 12) using time lapse photos at 10 min intervals. ** Day = Day of observation (days 1, 8, 10 and 12). *** NA: Treatment × Day interaction was non-significant and removed from the model.

**Table 3 animals-12-01768-t003:** Effects of enrichment treatment and day of observation on the postures of sows observed in photo scans. LS Means of duration (prop. of observations) and number of sows in each posture (prop. of sows).

Behavior *	Treatment				SEM	*p*-Value	
	Control	Constant	Rotate	Stimulus		Treatment	Day **
Standing							
Time (prop of obs.)	0.489	0.542	0.589	0.563	0.033	0.233	0.071
Number of sows (prop)	0.053 ^a^	0.063 ^b^	0.065 ^b^	0.067 ^b^	0.002	0.003	0.155
Sitting							
Time (prop. of obs.)	0.168	0.125	0.118	0.103	0.025	0.295	0.287
No. of sows (prop)	0.038	0.043	0.041	0.041	0.003	0.620	0.779
Lying							
Time (prop of obs.)	0.950	0.949	0.954	0.955	0.011	0.979	0.661
No.of sows (prop)	0.128	0.105	0.137	0.118	0.007	0.061	0.616

* Sows were observed during an 8-h period (8:00–16:00 h) on four days (1, 8, 10 and 12) using time lapse photos at 10-min intervals. ** Day = Day of observation (days 1, 8, 10 and 12). ^ab^ LS Means with a different superscript within the same row are significantly different (*p* < 0.05).

**Table 4 animals-12-01768-t004:** Effects of social status and day of observation on time spent near and in contact with enrichment (prop. of observations) and number of sows near or in contact with enrichment (prop. of sows).

Behavior *	Social Status **		SEM	*p*-Value ***	
	Dom	Sub		SS	Day
Enrichment contact					
Time (prop. of obs.)	0.114	0.096	0.014	0.139	0.009
Number of sows (prop.)	0.353	0.355	0.004	0.882	0.015
Less than 1 M					
Time (prop. of obs.)	0.072	0.096	0.017	0.276	0.001
Number of sows (prop.)	0.345	0.344	0.006	0.799	0.098
Greater than 1 M					
Time (prop. of obs.)	0.514	0.487	0.043	0.646	0.008
Number of sows (prop.)	0.386	0.395	0.009	0.430	0.588

* Sows were observed during an 8-h period (8:00–16:00 h) on four days (1, 8, 10 and 12) using time lapse photos at 10-min intervals. ** Social Status LS Means: Dom = Dominant; Sub = Subordinate. *** SS = Social Status, Day = Day of observation (1, 8, 10 and 12).

**Table 5 animals-12-01768-t005:** Effects of enrichment treatment on time focal sows spent in contact with enrichment, less than 1 M or greater than 1 M from enrichment (% of observations) and number of focal sows near or in contact with enrichment (% of focal sows).

Behavior *	Treatment **			SEM	*p*-Value
	Constant	Rotate	Stimulus		Treatment
Enrichment contact					
Time (prop. of obs.)	0.065 ^a^	0.144 ^b^	0.132 ^b^	0.002	0.001
Number of sows (prop.)	0.341 ^a^	0.357 ^b^	0.363 ^b^	0.004	0.019
Less than 1 M					
Time (prop. of obs.)	0.045 ^a^	0.130 ^b^	0.105 ^b^	0.022	0.161
Number of sows (prop.)	0.334	0.361	0.340	0.008	0.161
Greater than 1 M					
Time (prop. of obs.)	0.623 ^a^	0.441 ^b^	0.436 ^b^	0.055	0.033
Number of sows (prop.)	0.399	0.389	0.384	0.011	0.588

^a,b^ LS Means with a different superscript within the same row are significantly different (*p* < 0.05). * Sows were observed during an 8-h period (8:00–16:00 h) on four days (1, 8, 10 and 12) using time lapse photos at 10-min intervals. ** LS Means of proportions.

**Table 6 animals-12-01768-t006:** Effects of social status, enrichment treatment and day on focal sow postures observed in photo scans. LS Means of duration (% of observations) and number of sows in each posture (% of sows).

Behavior *	Social Status **		SEM	*p*-Value		
	Dom	Sub		SS	Treatment	Day
Standing						
Time (prop. of obs.)	0.203	0.186	0.013	0.324	0.125	0.104
Number of sows (prop.)	0.371	0.382	0.005	0.163	0.501	0.038
Sitting						
Time (prop. of obs.)	0.058 ^a^	0.041 ^b^	0.004	0.014	0.643	0.563
Number of sows (prop.)	0.342	0.333	0.003	0.058	0.097	0.072
Lying						
Time (prop. of obs.)	0.499	0.510	0.032	0.805	0.748	0.509
Number of sows (prop.)	0.383	0.393	0.007	0.304	0.675	0.768

^a,b^ LS Means with a different superscript within the same row are significantly different (*p* < 0.05). * Sows were observed during an 8-h period (8:00–16:00 h) on four days (1, 8, 10 and 12) using time lapse photos of 10-min intervals. ** Social Status LS Means: Dom = Dominant; Sub = Subordinate.

**Table 7 animals-12-01768-t007:** Effects of enrichment treatments and social status on skin lesion scores in focal sows. Lesion scores were measured on treatment days 1 and 12.

Lesion Scores *	Treatments				Social Status					
	Control	Constant	Rotate	Stimulus	SEM	*p*-Value	Dom	Sub	SEM	*p*-Value
Day 1 Total	0.19	0.62	0.26	0.33	0.44	0.065	0.28	0.32	0.28	0.602
Shoulder Day 1	0.17 ^a^	0.31 ^b^	0.07 ^c^	0.16 ^a^	0.31	0.011	0.17	0.13	0.22	0.396
Side Day 1	0.02	0.12	0.04	0.06	0.54	0.099	0.04	0.05	0.38	0.580
Day 12 Total	0.47 ^a^	0.21 ^b^	0.32 ^c^	0.77 ^d^	0.33	0.027	0.39	0.39	0.25	0.977
Shoulder Day 12	0.07	0.06	0.06	0.11	0.40	0.589	0.09	0.06	0.31	0.385
Side Day 12 **							0.03	0.04	0.46	0.575

^a,b,c,d^ Means with different superscript within same row are significantly different at *p* < 0.05. * Lesion scores were done using a scale of zero to three (0 = no injury, and 3 = severe injury) on 11 regions on the right and left side of sows. Total = sum of lesions for all 22 body regions. ** Side lesions on day 12: models of treatment effect did not converge due to the low number of lesions observed.

**Table 8 animals-12-01768-t008:** Saliva cortisol levels (µg/dL) in Dom and Sub sows.

Factor	Cortisol (µg/dL) **	SEM	*p*-Value
Social status *			
Dom	0.187	0.034	0.953
Sub	0.190	0.033	
Stage of gestation			
Early gestation (week 5)	0.133 ^b^	0.042	0.007
Middle gestation (week 9)	0.136 ^b^	0.042	
Late gestation (week 14)	0.298 ^a^	0.042	

^a,b^ Least square means with different superscript within a column are significantly different at *p* < 0.05. * Dom = Dominant (high ranking sows), Sub = Subordinate (low ranking sows), SEM; Standard error of means. ** Cortisol levels were determined from saliva samples collected from focal sows between 8 and 9 a.m. at three-time points: 5th week of gestation (early), 9th week of gestation (middle) and 14th week of gestation (late).

## Data Availability

The data presented in this study are available on request from the corresponding author.

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
