# Peer review of "Effects of Enrichment Type, Presentation and Social Status on Enrichment Use and Behavior of Sows—Part 2: Free Access Stall Feeding"

_animals, 2022, doi:10.3390/ani12141768_

Round 1

Reviewer 1 Report

The study investigates how enrichment use and habituation are influenced by a  Free Access Stalls (FAS) feeding system. The authors hypothesized that enrichment use in FAS would differ from Electronic Sow Feeding (ESF) by reducing social stress and aggression among sows. Apparently, the current study is a follow-up of a previous study by Roy et al. who evaluated the same treatments and variables in an ESF system.

The literature review is very good and it shows the state of the art of the theme. The results and discussion are appropriate.

I made a few comments in the annexed file.

Reviewer 2 Report

Really thorough methodology. In the results, I don't see an area which uses your data for parity code. When you discuss age of sows (part 3.2) are you referring to the code parity or the actual parity? The numbers imply actual parity. 

Here are some line edits I found while reading:

line 52: "o" should be "of" 

line 57- a little unclear. Possibly add "the lack of use of substrate" 

line 67- the first 'and' should be 'an' 

line 138- remove "sows" at the end of the line 

line 190- missing close of )

line 437- "paced" should be "placed"

Reviewer 3 Report

In the whole article, there are also some article usage problems of “the, a, or an”, and the comma usage problem. Also, the nouns may not agree in number with other words in the phrase. The problems above are not pointed out one by one in the reviewer’s suggestion. For example,

Line 15:  “provision” to “the provision”

Line 48:  “in North American farms ” to “on North American farms”

Line 50:  “the interaction” to “interaction”

Line 58:  “floor” to “floors”

Line 61:  “chains and even garden hoses” to “chains, and even garden hoses”

Line115- 116: How to place 8 replicates in four gestation pens?

Line151-158: You describe that “…chain (3 per pen)” in each treatment (pen), but there are only two “+” signs shown in Figure 1 as the enrichment position, how do you place 3 at the two positions?

In Line191-192 “Enrichment use was studied for the whole group and for the 6 selected sows (Dom and Sub) using the digital photos.” Does the whole group contain the 6 selected sows? Why do you emphasize the 6 selected sows here? n=28? And, in Line 187: You state that “taking into account only those sows that were observed clearly.” If the 6 selected sows could not be observed clearly, how do you deal with that? Besides, how many sows could be observed clearly and taken into account for the observation? (n=? in 28).

According to your results, the Rotate and Stimulus treatments showed completely similar behaviour performance which reveals that the different treatments of these two groups do not affect the sows. It seems the rotation of three enrichments in the Stimulus treatment (as described for Rotate) with an associative stimulus used to signal the arrival of enrichment could not access the aim to make the sows establish the relationship between the rotation and the signal stimulus. So, I doubt the rationality of the experiment design. I think there is no difference between these two groups. According to that, I think it is necessary to delete the inappropriate treatments and rewrite the article before publishing.

Round 2

Reviewer 3 Report

I think the author's answer is not enough to explain the question I raised, and the question part of the text has not been sufficiently scientifically explained and revised, so I insist that the current writing presentation of the article cannot meet the publication needs of the journal.